# An Innovative Approach: The Usage of N-Acetylcysteine in the Therapy of Pneumonia in Neonatal Calves

**DOI:** 10.3390/ani14192852

**Published:** 2024-10-03

**Authors:** Milan Ninković, Jadranka Žutić, Aleksandra Tasić, Sveta Arsić, Jovan Bojkovski, Nemanja Zdravković

**Affiliations:** 1Scientific Institute of Veterinary Medicine of Serbia, Janisa Janulisa 14, 11000 Belgrade, Serbiaalekstasic79@gmail.com (A.T.);; 2Faculty of Veterinary Medicine, University of Belgrade, Bulevar Oslobodenja 18, 11000 Belgrade, Serbia

**Keywords:** calves, N-acetyl cysteine, pneumonia, therapy

## Abstract

**Simple Summary:**

Treatment of neonatal pneumonia with N-acetyl cysteine (NAC) in calves shortens the time to symptom resolution by 27 h. There is a lack of data on NAC usage for neonatal pneumonia in calves. However, previous research explored its usage in other animals with other indications in cattle. The effects of NAC on neonatal pneumonia in calves were investigated in 40 animals from different owners who accepted or refused the addition of NAC in the therapy protocol.

**Abstract:**

NAC has mucolytic, antioxidant, and antimicrobial effects in living organisms. However, the therapeutic effects of NAC on clinical recovery among neonatal calves with respiratory diseases have not yet been studied. Our study represents the first investigation of the effects of NAC in neonatal calves with pneumonia. The objective of this work was to observe the effects of NAC in the treatment of neonatal pneumonia, including its ability to reduce the clinical score, shorten the duration of the treatment, and improve the overall health condition of neonatal calves. For this study, calves were divided into two groups: a treatment group that received NAC and amoxicillin with clavulanic acid, and a control group that received amoxicillin with clavulanic acid (antimicrobial only). The findings of this study indicate that NAC treatment significantly shortened the time to resolution (*p* < 0.001), compared to the results in the group without NAC treatment. Generally, NAC-supplemented therapy reduced the recovery time by more than 27 h (or slightly more than one day), compared to that in the antimicrobial-only group. Our study presents the first reported usage of NAC in therapy for respiratory disorders.

## 1. Introduction

Calf pneumonia can occur at any time during calfhood and has a multifactorial etiology [1,2]. The bovine respiratory disease complex in neonatal calves has an estimated financial cost per case of USD 42.15, including the use of anti-inflammatory medications in the treatment protocol [3]. Moreover, broad-spectrum antimicrobials are still widely administered to sick calves as a first-line response [4]. Furthermore, respiratory problems associated with dystocia may be lethal for neonatal calves due to the multifactorial etiology of relevant diseases [5]. Numerous factors contribute to dystocia and require assistance during calving. For example, manipulation of the fetus in the birth canal to reposition the fetus can lead to premature rupture of the umbilical vessels. In cattle, the umbilical cord is very short, increasing the potential of prematurely rupturing the umbilical cord in the birth canal and starting the breathing process. This process can produce aspiration of amniotic fluid, leading to aspiration pneumonia or even death depending on the time spent in the birth canal [6,7]. Perinatal asphyxia in calves leads to death of the body in the first hours of life. The length and method of delivery for dairy cows affect the immune profile of the newborn calf and may predispose the calf to certain diseases or conditions as a result of a difficult delivery [8]. The effects of dystocia on calves are trauma hypoxemia and failure of the transfer of passive immunity, which plays a critical role in the calf’s resistance to infection and its ability to maintain good health [7]. Calves born via prolonged calving are usually exhausted and ingest less colostrum; this may be the result of failure to suckle in time and, consequently, leads to the failure of the passive transfer of immunoglobulins [9]. Aspiration pneumonia most often occurs due to the aspiration of liquid in the lumen of the respiratory tract [10]. The presence of aspirated contents can seriously impair the animal’s respiratory functions, providing a suitable medium for the multiplication of respiratory syndrome pathogens, most of which are bacteria [10,11,12]. Several reports have described the use of different antimicrobials in the treatment of pneumoniae in calves [11,12]. Acetylcysteine belongs to a group of drugs called mucolytics, which have demonstrated antioxidant, antimicrobial, and anti-inflammatory properties in multiple models of disease in both human and veterinary medicine [13,14]. In humans, NAC is one of the most widely used therapeutic agents in the therapeutic protocols for cystic fibrosis and liver diseases, including acetaminophen (paracetamol) poisoning, acute liver failure, and alcoholic liver diseases [14,15]. NAC can also be found as an over-the-counter dietary supplement for humans in countries such as the United States, Canada, and Australia [13]. In dairy cows, there have been reported studies on the use of NAC in therapies for reproductive disorders and metabolic diseases such as ketosis [16,17]. NAC’s mechanism of action is based on stimulating the synthesis of glutathione, a compound that helps fight free radicals, which cause inflammation, oxidative stress, and cell damage. The mucolytic effects of NAC disrupt disulfide bonds between mucin polymers, reducing the viscosity of mucus. Consequently, NAC is used in therapies for endometritis and metritis in cattle and horses [14,15,16,18]. NAC has also demonstrated antimicrobial properties against Gram-positive and Gram-negative bacteria, blocking biofilm formation [19]. The application of NAC increases average daily gains and food intake and improves the growth performance of neonates [20].

Acetylcysteine’s sulfhydryl groups may hydrolyze disulfide bonds within mucin, breaking down the oligomers and making the mucin less viscous, thereby decreasing the viscoelasticity of airway secretions and improving mucociliary clearance [21]. Breaking of the disulfide bonds in the cross-linked mucosal protein of NAC manifests accelerates the process of mucus elimination, leading animals to cough up contents from the respiratory tract, and reduces irritation of the respiratory tract. NAC also offers mucolytic properties and acts as an antioxidant in human chronic obstructive pulmonary disease (COPD) [22].

Numerous studies have shown the therapeutic effects of numerous antibiotics used to treat pneumonia in calves [10,23,24,25]. Numerous antibiotics employed in therapies for bovine pneumonia, such as tetracyclines, macrolides, β-lactam antibiotics, cephalosporins, and fluoroquinolones, are frequently utilized to combat BRD, with long-acting formulations ensuring ease of use in feedlot settings. In addition to antibiotics, glucocorticosteroids are often used in the treatment of pneumonia, although their use remains controversial. The synergistic effects of NAC with other β-lactam antibiotics have also been demonstrated in the treatment of bovine mastitis, modulating antibiotic susceptibility against the pathogens [26]. However, the therapeutic effects of NAC on clinical recovery in neonatal calves with respiratory diseases have not yet been explored. We hypothesized that the intramuscular application of NAC may reduce the clinical score, shorten the duration of the treatment, and improve the health conditions of neonatal calves.

## 2. Materials and Methods

This study was conducted in compliance with Serbian law on animal welfare (Official Gazette of the Republic of Serbia No 41/09) and the related Ordinance (Official Gazette of the Republic of Serbia No 39/10).

### 2.1. Animals and Management

This study included a total of 40 Simmental calves, 0–3 days old, with respiratory signs. All included calves had a history of dystocia. All the calves were born through the birth canal and no calf born via c-section was included in the study. This study was performed from September 2023 to February 2024, in Macva County, Serbia, using calves from four dairy farms with capacities ranging from 15 to 75 dairy cows. The husbandry conditions for calf holding and feeding were similar on all included farms. Considering the owner’s choice of therapy, neonatal calves (*n* = 40) with respiratory impairments were placed into two study groups based on the diagnosis, and these calves were evaluated and monitored during the therapy.

### 2.2. Experimental Design

Calves were selected based on Wisconsin clinical respiratory scoring and the work of Vermorel et al. on the clinical findings of elevated body temperature with hyperthermia (>39.5 °C) [27]. The calves were placed into two groups. The calves were then offered standard treatment (Atb) or extended treatment (Atb + NAC); some farmers accepted the treatment, whereas some declined due to the increased cost associated with the addition of NAC. All neonatal calves were treated intramuscularly using amoxicillin with clavulanic acid, but the calves whose farmers agreed to the extended treatment received an additional intramuscular application of NAC.

Treatment group: *n* = 20 calves treated using 6 mg of acetylcysteine with amoxicillin and clavulanic acid intramuscularly once daily on the side of the neck based on 8.75 mg/kg bodyweight (7.0 mg amoxicillin, 1.75 mg clavulanic acid), equivalent to 1 mL of suspension per 20 kg bodyweight (Synulox RTU).

Antimicrobial-only group: *n* = 20 calves treated using amoxicillin with clavulanic acid intramuscularly administered once daily on the side of the neck based on 8.75 mg/kg bodyweight (7.0 mg amoxicillin, 1.75 mg clavulanic acid), equivalent to 1 mL of suspension per 20 kg of bodyweight (Synulox RTU).

Treatment in the groups was continued once daily until symptoms disappeared. N-acetylcysteine (Zambon s.p.a., Milano, Italy) was injected intramuscularly with a zero-day meat withdrawal period.

### 2.3. Clinical Scoring Respiratory System

All calves included in this study were rated using the Wisconsin clinical score with the addition of lung auscultation. The Wisconsin clinical score includes five clinical parameters (nasal discharge, ocular discharge, rectal temperature, cough, and ear position). Each category was assigned scores ranging from 0 to 3 as the clinical signs [28]. The calves’ health was checked twice a day.

All calves were considered to have respiratory disease according to relevant clinical criteria [29]. Calves presenting the following features were considered recovered: no pneumonia, temperature within a normal range, normal respiratory frequency, satisfactory appetite, unimpaired general health, no increased bronchial tones, slightly thin nasal discharge, and rare soft coughing. The respiratory system was assessed via a visual inspection (respiratory rate, nasal discharge, type of breathing, shortness of breath, and dry or wet spontaneous cough) and auscultation (increased or decreased volume of breathing sounds, rales, and signs of breathing difficulty). The classic clinical presentation of an affected calf with respiratory symptoms included fever, loss of appetite, lethargy, and respiratory distress including nasal discharge, cough, with nasal and eye discharge, rapid breathing (tachypnea), excessive mucus secretion, difficulty breathing (dyspnea), and open-mouthed breathing. Calves were auscultated to determine the presence of any abnormal lung sounds including crackles and wheezing [30].

Calves followed a morning–evening schedule during the treatment period. The recovery rates of calves were described as the percentage of calves with no signs of respiratory impairment according to a clinical score < 5 [27].

### 2.4. Statistical Analysis

Statistical analyses included a descriptive statistical analysis, followed by a comparison between the two groups using a Mann–Whitney U-test. Linear regression analysis with the backward method was applied to measure the assessed variables for the clinical resolution periods. Data analyses were performed in JASP (ver 0.18.3.0, JaspTeam, Amsterdam, The Netherlands).

## 3. Results

All 40 calves featured similar respiratory impairment, and the results of the initial screening showed invariable parameter values. The results of the clinical scores between the compared groups before treatments are provided (Appendix A). Both groups presented no statistical differences in the signs’ grades (Appendix A). No calf deaths were recorded in any of the studied groups during therapy.

The therapy protocol including NAC yielded a faster recovery time than that in the antimicrobial only. The average clinical recovery time in the Atb+NAC group was calculated to be 41.4 ± 12.6 h (41 h and 24 min ± 12 h and 36 min), while that of the antimicrobial-only group was 72.6 ± 25.07 h (72 h and 36 min ± 25 h and 4 min) (Figure 1). The expressed difference in time to resolution was statistically significant *p* < 0.001 (W = 352) between the Atb+NAC and Atb groups. In plain language, NAC reduced the length of therapy by one day.

We then formed a linear regression model to further examine the effects of additional NAC treatment on neonatal calves in need. The initial linear regression model is presented in Appendix A. This analysis shows the expected recovery time (in hours) according to the clinical parameters, with the final model indicating the importance of the treatment group and rectal temperature as parameters for predicting time to resolution (Table 1).

The final regression equation explains 84.66% of the symptoms’ time to resolution and includes the intercept coefficient, group (0 for the Atb group and 1 for the Atb+NAC (supplemented treatment), and rectal temperature (°C), which is further explained in Equation (1):Time of symptom resolution = −1258.41 − 27.74 Group + 32.91 Temperature (°C).(1)

The results show that the inclusion of NAC treatment in therapy reduced the recovery time by 27.74 h (27 h and 44 min), which is slightly more than one day. This result suggests that a clinically evident sick period can be shortened if farmers select a comprehensive treatment option. An additional predictive parameter was rectal temperature, as time to resolution was prolonged by 32.91 h (32 h and 54 min) with each increment of 1 °C.

## 4. Discussion

To the authors’ best knowledge, this research presents the first report on the application of NAC in therapy for respiratory diseases among calves. NAC was previously used in the treatment of mastitis and endometritis in cows [16,26]. The prevalence of the bovine respiratory disease complex affecting pre-weaned dairy calves ranges within-herd prevalence varied from 0 to 22% [31]. The crucial success factor for pneumonia therapy in calves is treatment during the early onset of symptoms as quickly as possible to reduce the possible long-term pulmonary damage with pneumonia development [32,33]. Early veterinary intervention guarantees immediate attention to potentially lethal conditions and ensures the responsible use of antibiotics for more severe clinical signs [34]. Standard protocols for the treatment of respiratory diseases in cattle are based on the use of antibiotics, in addition to NSAID drugs, which play a supportive role in therapy and exhibit anti-inflammatory and antipyretic effects. Corticosteroids, rehydration agents, and vitamins are also used in therapies for respiratory diseases [35]. In the first hours of life, newborn calves must consume a sufficient amount of colostrum, which is necessary for health, vitality, and adequate passive immune protection [36]. As calves are born agammaglobunemic, the passive transfer of colostrum from a cow is critical for newborns, especially as pneumonia can occur almost immediately at birth. Therefore, pneumonia management must be available during the entire period from birth to weaning [37]. Passing through the birth canal, the calf encounters different bacterial flora. Due to its long stay in the birth canal in the cases of assisted calving, a calf can suffer the aspiration of amniotic fluid, which is a medium that facilitates the multiplication of bacteria and initiates inflammatory processes in the lungs of newborn calves [10]. Pneumonia in newborn calves has a tendency to develop and spread quickly, making it necessary to apply adequate symptomatic therapy [7]. Hyperemia can occur in the early stages of pneumonia. This disorder refers to the appearance of edema in the bronchoalveolar pathways with increased permeability of blood vessels, which results in the release of fibrin into the lumen of the alveoli, reducing the respiratory surface. TNF macrophages and neutrophil granulocytes accumulate due to the production of pro-inflammatory leukotrienes IL-1, IL-6, and IL-8. The release of inflammatory mediators leads to hypoventilation, the presence of edema, a decrease in the respiratory surface, and the removal of surface surfactants. These processes produce an increase in surface tension followed by expiratory dyspnea in sick calves. The presence of proteins in the resulting exudate complicates the process, leading to the formation of a hyaline membrane that prevents respiration [38,39]. The presence of liquid in the lumen of the alveoli prevents adequate respiration, which results in the appearance of cyanosis [8]. Auxiliary measures, such as the use of NSAID drugs, are of great importance in therapy for pneumonia in calves to reduce the symptoms of the disease [40]. However, there are too few studies investigating the use of additional supportive drugs, such as mucolytics or bronchodilators, in the treatment of pneumonia in cattle. In humans, NAC appears to be a promising therapeutic agent in the treatment of pulmonary diseases [14]. Based on the approval of the Commission for Veterinary Medical Products, this drug is approved for use in all kinds of animals intended for the production of food for humans. NAC is characterized by good bioavailability in all tissues and organs and very quickly achieves therapeutic effects similar to those seen in humans within 1–4 h of application, with the lowest activity found in the muscles and brain [41]. Based on the toxicological findings for acetyl cysteine, the minimal risk levels established are not important because residues of this antioxidant do not have a harmful effect on human health. Ultimately, NAC offers low toxicity for both animals and humans [42].

The application of NAC is also becoming more common in therapies for various pathological conditions in other animals, such as weaning disorders in piglets, porcine epidemic diarrhea, and chronic obstructive pulmonary disease in horses [18,43,44]. NAC’s mechanism of activity in the body is based on its mucolytic, antioxidant, anti-inflammatory, and antimicrobial properties, which provide a wide range of indications for use [13]. The protective effects of NAC against oxidative stress caused by various inflammatory processes are based on the prevention of lipid peroxidation and the release of free radicals, preventing cell damage [43]. The antioxidant effects of NAC have been used to prevent reproductive disorders during oocyte maturation, increase the blastocyst rate, promote early embryo formation, improve the success of early embryo development, and reduce embryo losses [45]. Moreover, maintaining the antioxidant status of goats can improve the quality of milk and the health status of milking goats. NAC has positive antioxidant effects with a positive regulatory effect on the reproductive function activation of the PI3K/Akt signaling pathway. In addition, the altered expression levels of the CSF-1, BDNF, WIF1, ESR2, TGF-β3, CTSS, and PTX3 genes may be important mechanisms through which NAC regulates the prevention of reproductive disorders in the early stages of pregnancy in goats [46].

In our study, NAC accelerated the healing process and reduced the treatment time. The cough duration was shortened, and quicker elimination of mucus from the respiratory tract with a normalized breathing rhythm was noted compared to the antimicrobial-only treatment group.

Unlike our results, this previous study used a longer therapy with the application of amoxicillin with clavulanic acid together with chlorpheniramine and meloxicam injections, with clinical recovery lasting for 5 days [26]. The use of enrofloxacine with dexamethasone and turpentine oil inhalation cured the disease after 4–5 days, with a recovery rate of 91% [45]. Besides NAC, other medications reported for the treatment of calves with respiratory distress syndrome include nebulized salbutamol, fluticasone, and furosemide [32]. The synergistic antimicrobial effects of NAC and penicillin antibiotics were also previously reported [20]. According to our results, the use of NAC in combination with β-lactam antibiotics shortens the length of therapy and improves therapeutic responses compared to treatments using only antibiotics. Due to the continuous growth of antimicrobial resistance, when choosing antibiotics for the treatment of pneumonia, antibiotics from an older group of drugs should be chosen first, followed by newly synthesized drugs, which should only be used as an extension of therapy if the first choice fails [35,47].

NAC is approved for use in farm animals as it does not leave residues in meat and milk [42,48]. In horses and mares, NAC was found to be safe under intravenous, oral, and intrauterine administration [18]. For NAC in animal therapy, there are insufficient data on the method of application, such as the dosing schedule and the dose itself. However, in humans, NAC has a wide dosage interval depending on the indication. The largest dose is used in acetaminophen poisoning, for which NAC is an antidote if applied at a dose of 150 mg/kg bolus and 12.5 mg/kg/hour infusion for a total of 400 mg/kg [14]. Dose titration was not performed for this study due to the wide therapeutic index of NAC, but we recommend future research defining the recommended dose of NAC for calves and other categories of cattle.

The application of NAC therapy reduced the levels of pro-inflammatory cytokines (IL-1β, Il-8, and TNF-α) and oxidative modifications in lung tissue and improved lung function parameters [49]. The antioxidant activity of NAC is based on the ability of its free thiol group to react with reactive oxygen and nitrogen species generated during inflammatory processes [13], which may explain the decreased duration of therapy observed among the calves.

The more rapid clinical recovery and reset of the Wisconsin clinical respiratory score parameters from ill to physiological is the most obvious consequence of NAC, presumably due to anti-inflammatory activities, as NAC was previously found to block the initiation of inflammatory cytokines, exert strong antioxidant functions, promote the production of lung surfactants, protect alveolar elasticity, and thus improve lung function [14].

In cows, oral and intravenous NAC supplementation leads to improved health, antioxidant capacity, and inflammatory responses and also increases the capacity of antioxidant enzymes in the body [50]. Dietary supplementation of NAC, for example, was found to increase the activity of the endogenous antioxidants superoxide dismutase (SOD), catalase (CAT), and glutathione peroxidase (GSH-Px) [13,50]. NAC application was also found to have an immunomodulatory effect under stressful conditions, improving the number of leukocytes, neutrophils, and monocytes, thereby enhancing immune responses [14,51]. The transition from milk to concentrated nutrition places significant stress on sucklings, and the oxidative stress induced by weaning promotes enteral villus atrophy and suppresses the activities of digestive enzymes in weaned piglets [52]. NAC has beneficial impacts on the number of probiotic bacteria in the intestine: *Lactobacillus* sp. and *Bifidobacterium* sp. counts were increased, while *E. coli* counts were reduced, presumably due to NAC’s antioxidant properties. Moreover, the lactobacilli and bifidobacteria counts were positively correlated with the activities of antioxidant enzymes [43]. The positive effect of NAC on the development of the intestinal flora was also confirmed in piglets [5]. The application of NAC increased the presence of the beneficial microflora *Megasphaera*, *Lactobacillus reuteri*, and *Megasphaera elsdenii* and reduced the abundance of *Phascolarctobacterium succinatutens*, *Prevotella copri*, and *Selenomonas bovis*, stimulating the immune response and preventing the occurrence of diarrhea [52].

Even with older animals, the rectal temperature is a very crucial clinical finding. In young bulls with a spontaneous bovine respiratory disease complex, a mean temperature of 40.1 °C was accompanied by clinical signs such as depression, cough, nasal discharge, and occasional ocular discharge [31,53]. Rectal temperatures were previously recognized as the most prominent diagnostic tool with exceptional accuracy, especially in the first 24–72 h after disease onset [30]. Our results showed temperature to be a significant factor in predicting the duration of respiratory symptoms. A case of calf pneumonia is defined as distress, inducible cough, and abnormal respiratory auscultation followed by a body temperature > 39.5 °C without other fever development [54].

In several studies, the antipyretic effects of NAC were based on the IL-10 pathway, which blocks the production of pyrogenic cytokines TNF-α, IL-1β, and IL-6 [55,56]. The above data support our results showing the NAC treatment group to have a faster clinical recovery, which confirms the antipyretic effects of NAC in the treated calves. However, we did not use additional antipyretic agents. Additionally, the antipyretic effects of NAC may be due to the modulation of transcriptional activities through several pathways involving nuclear factor-κB (NF-Κb) and cyclin inhibitors [57].

As a potential auxiliary antiviral treatment, NAC was found to prevent the occurrence of porcine epidemic diarrhea virus. We assume that cellular immune response stimulation invokes a response from the T-lymph node cells and strengthens the adhesion and phagocytosis of macrophages, thereby protecting cells from virus replication [58].

The administration of NAC may block hepatic lipid accumulation and provide therapeutic benefits against the metabolic complications found in fatty liver disease due to antioxidant effects and the prevention of fat peroxidation [59]. The antioxidant role of NAC was also described in cows with ketosis [60]. Neurohormonal changes occur in the peripartum periods of highly lactating cows, which results in uncontrolled lipomobilization and the occurrence of fatty liver syndrome. Due to the high concentration of ketone bodies and beta-hydroxybutyric acid, free radicals are produced, leading to a decrease in lipid peroxidation capacity and damaged hepatocytes due to the inability of the mitochondria to oxidize fatty acids [59].

The survival mechanisms of some bacteria that cause chronic infections, such as S. aureus and *P. aeruginosa*, are based on biofilm production, consequently increased resistance to host defenses, and drastically reduced sensitivity to antimicrobial agents. Due to its multifactorial nature, NAC was also described as a preventive biofilm formation agent, representing an open field for future research [61].

The local farmers that declined the NAC treatment had empirical knowledge of neonatal pneumonia and believed that the additional therapy cost would be greater, without anticipating the shorter recovery time. As the observed calves originated from several farms, the farm effect on recovery could be important. To reduce eventual bias due to a farm/calf carer effect, all calves were seen twice a day. The most significant finding of this study is the effectiveness of NAC in treating respiratory disorders among calves with a shorter time to resolution. However, the economic effects of common diseases among contemporary group-housed calves are little known among small farmers. According to economic calculations in local circumstances, using NAC in calf therapy saves about EUR 16 per calf, considering one fewer veterinarian visit due to the shortened recovery period and the additional projected cost of NAC. Although we did not observe any deaths in our study, the projected economic losses due to calf deaths from respiratory disease may reach up to USD 845 [62]. It was previously reported that a single dairy calf pneumonia case has a cost of GBP 43.26 per animal, with estimated costs rising to GBP 82.10 for suckler calves and GBP 74.10 for older calves [63]. Finally, these estimates do not consider the costs of calf welfare or farmers’ emotional costs [34].

## 5. Conclusions

The administration of NAC alongside standard antibiotic therapy seems to promote the faster recovery of affected calves, thereby decreasing the duration of treatment. This study filled a research gap by applying NAC to treat calf respiratory disorders. The results showed a significant reduction in recovery time among the group of calves treated with NAC. While already available for cows, horses, and swine, adequate wide-spectrum NAC dosage regimes and treatment protocols for calves should be developed as a promising agent facilitating the efficient treatment of respiratory diseases.

Based on all the above, the application of NAC represents an assuring choice for therapeutic and/or preventive applications to resolve various disorders, bearing in mind the antioxidant’s wide availability and polyvalent role in animals.

## Figures and Tables

**Figure 1 animals-14-02852-f001:**
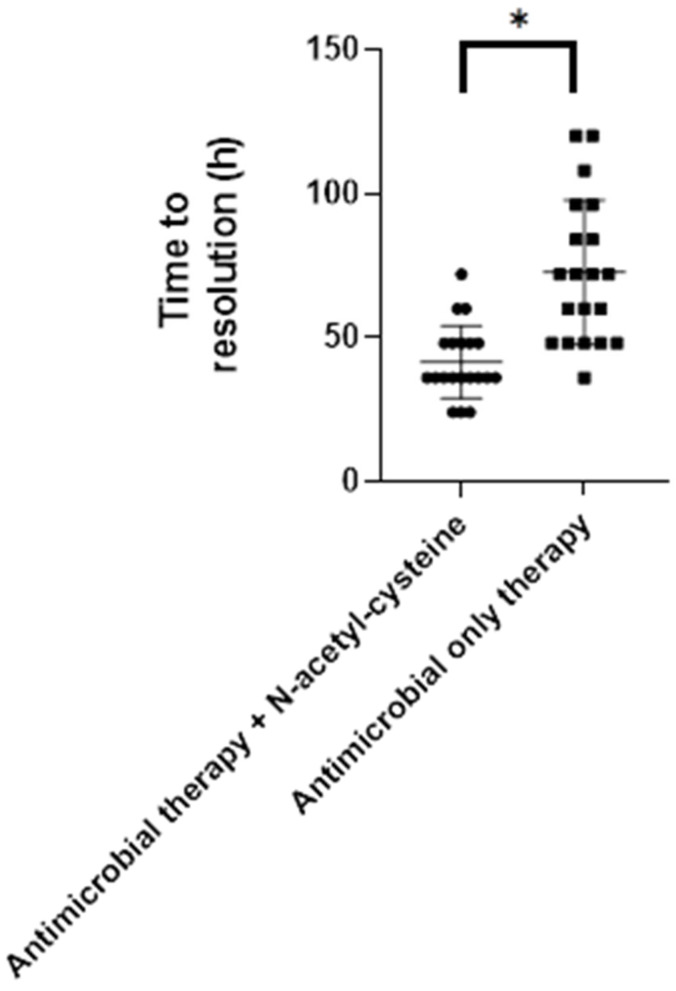
Calves with N-acetyl-cysteine treatment supplemented antimicrobial therapy (dots) presented a significantly shorter time to resolution (asterisk presents statistically significant difference *p* < 0.001) than the group antimicrobial-only therapy (squares).

**Table 1 animals-14-02852-t001:** Final linear regression model.

*n* = 40	Regression Summary for Dependent Variables: Time Resolution (Group 0 i 1) R = 0.92012484 R^2^ = 0.84662972 Adjusted R^2^ = 0.83833944 F(2,37) = 102.12 p					
	b *	Std. Err. of b *	b	Std. Err. of b	*t* (37)	*p*-Value
Intercept			−1258.41	127.42	−9.88	<0.00
Group	−0.56	0.06	−27.74	3.22	−8.63	<0.00
Temp (C)	0.68	0.06	32.91	3.15	10.45	<0.00

b *—normalized β coefficient.

## Data Availability

Data available on request from the authors.

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
