# Peer review of "An Innovative Approach: The Usage of N-Acetylcysteine in the Therapy of Pneumonia in Neonatal Calves"

_animals, 2024, doi:10.3390/ani14192852_

Round 1

Reviewer 1 Report

Comments and Suggestions for Authors

My comments and suggestions are in the attached report.

Comments on the Quality of English Language

There are too many sentences that are difficult to understand mostly due to incorrect ordering of words or similar.

Author Response

Dear reviewer

Firstly, we would like to thank you for your comments, as we see the way that they greatly improve our work. Some of your comments inspired nice discussion among authors it made us think twice so we fill that we have been learning form you and your comments. We tried to give adequate answers to your suggestions. We apologize for the errors in the references to our document, the references are adequate as stated. We have listed the references individually according to your requests in the answers. The final version of the manuscript was revised over MDPI language editing service.

Comments

Abstract:

Line 10: Sentence requires to be rewritten.

Response: English corrected.

Line 15: Presumably, NAC does not have an anti-inflammatory effect on bacteria.

Response: Thank you for comment. We have corrected the mistake.

Line 16: The claim that pneumonia is the leading cause of death in neonatal calves is dubious. Trauma, anoxia, and septicaemia are all more likely. The claim must be referenced in the introduction.

Response: We have removed the sentence.

Line 18: Sentence needs to be more specific.

Response: We made a change.

Line 19: As above. Case selection appears to be on neonatal pneumonia only.

Response: We made a change.

Line 25: Sentence requires to be rewritten.

Response: English corrected.

Introduction:

Line 31: Sentence needs to be rewritten. Furthermore, the claims for “great economic costs” (refs 1 and 2) refer to bovine respiratory disease in general and not specifically to neonatal pneumonia. Neither reference 1 nor 2 is a study of the economic costs of calf respiratory disease.

Response: The claims for “great economic costs” have removed from sentence.

Line 32: The claim of the financial cost of the bovine respiratory disease complex in neonatal calves is not supported with a reference.

Response: We have performed a reference replacement.

Line 35: Sentence requires to be rewritten.

Response: English corrected.

Line 38: sentence requires to be rewritten.

Response: English corrected.

Line 39: Reference 4 is on the microbiome in pig gut and the effects of NAC. It is not a study on neonatal calf disease.

Response: Please see, reference reference 4: Vestweber, J. G. (1997). Respiratory problems of newborn calves. Vet. Clin. North Ame. Food Anim. Pract. 1997, 13(3), 411-424.

Lines 41 to 45: References 5 and 6 are not studies on the bovine birth process and the negative impact that dystocia has on calf survival.

Response: We have performed a reference replacement.

Wells, S. J., Dargatz, D. A., & Ott, S. L. (1996). Factors associated with mortality to 21 days of life in dairy heifers in the United States. Preventive Veterinary Medicine, 29(1), 9-19.

Lombard, J. E., Garry, F. B., Tomlinson, S. M., & Garber, L. P. (2007). Impacts of dystocia on health and survival of dairy calves. Journal of dairy science, 90(4), 1751-1760.

Line 46: This sentence on the impact of the duration of parturition etc. on the “immune profile” of the newborn calf and predisposition to neonatal diseases is incorrectly supported by a reference on the impact NAC has on antipyresis mechanisms (reference 7).

Response: Thank you, the reference was put in correct numeration sequence

 7.Sanchez-Salcedo, J., Bonilla-Jaime, H., Lozano, M. G., Hernandez-Arteaga, S., Greenwell-Beare, V., Vega- Manriquez, X., Jimenez-Collado, C., Orozco-Gregorio, H. Therapeutics of neonatal asphyxia in production animals: a review. Vet. Med.- Chech. 2019, 64(5), 191-203

Line 50: It is not clear why this statement is made. Furthermore, the reference given is one on the impact of NAC on activated macrophages (ref 8).

Response: Please check the first version of the manuscript with the accompanying references according to us this reference is adequate to the citations. Please see, reference 8. Shakespeare, A. S. Aspiration lung disorders in bovines: A case report and review. JSAVA. 2012, 83(1), 1-7.

Lines 51-54: How can aspirated contents provide a suitable medium for viruses? Once again, the references (6, 9, 10) are not studies on the pathogenesis of aspiration pneumonia.

Response: We have performed a reference replacement. We have modified the sentence and deleted the viruses.

Lines 54-57: Reference 12 is on the role of acetoacetate in the pathogenesis of liver damage in ketotic cows and not on the qualities of NAC in human or veterinary medicine.

Response: Thank you, we corrected this in reference 12 is review paper: Tieu, S., Charchoglyan, A., Paulsen, L., Wagter-Lesperance, L.C., Shandilya, U.K., Bridle, B.W., Mallard, B.A., Karrow, N.A. N-Acetylcysteine and Its Immunomodulatory Properties in Humans and Domesticated Animals. Antioxidants (Basel). 2023 Oct 16;12(10):1867. doi: 10.3390/antiox12101867.

Please see, the paper.

Lines 58-60: Neither reference 12 or 13 are on the role of NAC in the treatment of liver conditions in the human.

Response: Thank you, we corrected this. Please see reference.

  1. Tieu, S., Charchoglyan, A., Paulsen, L., Wagter-Lesperance, L.C., Shandilya, U.K., Bridle, B.W., Mallard, B.A., Karrow, N.A. N-Acetylcysteine and Its Immunomodulatory Properties in Humans and Domesticated Animals. Antioxidants (Basel). 2023 Oct 16;12(10):1867. doi: 10.3390/antiox12101867
  2. Mokra, D., Mokry, J., Barosova, R., Hanusrichterova, J. Advances in the Use of N-Acetylcysteine in Chronic Respiratory Diseases. Antioxidants, 2023, 12(9), 1713.

Line 60-62: Whether NAC can be purchased over the counter in specified countries is of dubious relevance to this study.

The accessibility of the drug in use is quite important in the picture of treatment, as it introduces the reader that farmers may or may not have access to the drug without the doctor. Authors opinion is that this information is of a value to reader due to our experience that farmers might use over the counter medicines on their own unaware of consequences. Luckely NAC is very safe, but some drugs are not.

Lines 62-64: Neither reference 14 nor 15 refer to work carried out in dairy cattle.

Response: Please see, check the reference refer to work carried out in dairy cattle.

Reference 14: Tras, B., Dinc, D. A., & Uney, K. The effect of N-acetylcysteine on the treatment of clinical endometritis and pregnancy rate in dairy cows. Eurasian J Vet Sci. 2014, 30(3), 133-137.

Reference 15: Wang, Y., Li, C., Ali, I., Li, L., Wang, G. N-acetylcysteine modulates non-esterified fatty acid-induced pyroptosis and inflam-mation in granulosa cells. Molecular Immun. 20 20, 127, 157-163

Line 64: The sentence requires to be rewritten. Presumably, the important point is that it is glutathione that is the active agent. It is important to have this section properly referenced with studies that have demonstrated each of the effects claimed for NAC.

Response: We added the reference.

Line 66: This sentence requires to be rewritten. The subject of reference 14 is not cattle metritis nor endometritis.

Response: Please see and check reference.

Reference 14: Tras, B., Dinc, D. A., Uney, K. The effect of N-acetylcysteine on the treatment of clinical endometritis and pregnancy rate in dairy cows. Eurasian J Vet Sci. 2014, 30(3), 133-137

Line 70: This sentence appears to be out of the context of the preceding or following discussion.

Response: We deleted the sentence.

Line 73: The claim that the study evaluated the mucolytic and anti-inflammatory activity of NAC calves cannot be supported as neither activity was measured.

Response: We have made changes to the sentence.

Line 74: The sentence needs to be rewritten.

Response: English corrected.

Material and Method

Lines 82: The breed of the calves is important in this study. There may be a difference in the risk and severity of dystocia in cross bred calves compared to pure bred calves (of Holstein and Simmental) due to the likely impact on calf birth weight (note that the heaviest calf in the study was 39% heavier than the smallest). The severity of dystocia and how it was assessed is also clearly of relevance. How was the dystocia identified and graded and by whom across the study farms? Also the impact of thoracic injury and of prolonged anoxia and acidosis on the respiratory system results in clinical signs that are difficult to distinguish from neonatal pneumonia.

Response: We added breed. Cows with dystocia required obstetrical assistance due to irregular situs, habitus, disproportion of the fetopelvis. Consequently, it leads to longer retention of the fetus inside the birth canal. All the calves were born through birth canal and we haven’t considered c-sectioned deliveries in this manuscript.

We have considered only dystocia history in deliveries as anamnestic data for calves, the obstetric procedures were done by several different veterinarians and specialistic services while the closer introduction to delivery pathology is not in the scope of our work, and I am afraid, that not all the doctors would be willing to share data.

Line 83: Incorrect use of the word “symptom” and it should be clinical sign or similar.

Response: We replaced with clinical sign.

Line 86: By the description given, the allocation of animals to the treatment group was neither random nor blind. Presumably as the farmer made the decision on inclusion in NAC treatment on cost this could be affected by the assessment of the value of the calf or the farmer’s assessment of likely recovery. However, the statement “where some farmers accepted the treatment, some declined due to the increased cost of the treatment.” This raises the possibility that all animals from a single farm fell into a single treatment group (not NAC). This needs to be clarified as the farm effect on recovery could be important.

Response: Thank you for comment. We added suggested sentence in lines 344-345 in discussion All farms have similar housing conditions and the same feeding method, they are within 200km circle therefore we don’t expect much influence of different farm management on the effect of therapy. But it certainly stands your point, and for some other observations the farm micro locations and management make a huge difference.

2.2 Experimental design

Lines 90-92: This sentence needs to be rewritten. It is necessary to state the score threshold and to briefly describe the scoring system. The sentence simply refers to “elevated body temperature with hyperthermia (>39.5 o C)”. Reference 18 does not describe the Wisconsin clinical respiratory coding system.

Response: This sentence summarized design of study. We added the reference 19 for Wisconsin clinical respiratory scoring system.

Lines 92-93: This states that cases were alerted to the local veterinarian for treatment. However, the precise role of the veterinarian is not described. Did they pass on the case to study veterinarians, or had they been trained in the study protocol?

Response: Principal investigator M.N and N.Z observed all selected calves in coordination with the local veterinarian contacted by the owners for therapy. Clinical examination and assessment performed according by Vandermeulen J, Bahr C, Johnston D, Earley B, Tullo E, Fontana I, Guarino M, Exadaktylos V, Berckmans D. Early recognition of bovine respiratory disease in calves using automated continuous monitoring of cough sounds. Comput Electron Agric. 2016 Nov 1;129:15-26. doi: 10.1016/j.compag.2016.07.014.

Line 99: The clinical monitoring and frequency are described in the paragraph below.

Response: We deleted the sentence.

Line 110-112: This sentence should be included in the paragraph on clinical scoring. Clinical cure rate is not the correct term.

Response: We have made changes to the reordering of paragraphs.

2.3 Clinical scoring respiratory system

Lines 118-119: This indicates that the clinical assessment was made once a day.

Response: Thank you for comment. We maked writing error and

Line 123: States that absence of increased bronchial tones was part of the assessment in relation to cure, however no data on this is presented.

Response:

Line 134: This implies that calves were examined twice a day. This contradicts lines 118-119 and should be clarified.

Response: We made a writing error. Calves were examined a twice a day.

There is no description of the individual or individuals carrying out the assessment and clinical scoring. This is important as the variability between individuals carrying out the scoring and the specificity of the Wisconsin scoring system has been described (see Buczinski et al Prev Vet Med (2015) 119:227-231 and Moller et al (2024) J Dairy Sci 107:1102-1109) and should be acknowledged and accommodated within the analysis of the results.

Response: Thank you for the recommended reference. Scoring and the specificity of the Wisconsin scoring system has been described by Vandermeulen J, Bahr C, Johnston D, Earley B, Tullo E, Fontana I, Guarino M, Exadaktylos V, Berckmans D. Early recognition of bovine respiratory disease in calves using automated continuous monitoring of cough sounds. Comput Electron Agric. 2016 Nov 1;129:15-26. doi: 10.1016/j.compag.2016.07.014.

2.4 Statistical Analysis

The use of the t test for time to resolution is questionable as the distribution is likely to be skewed towards short recovery time. The twice a day assessment (12 hourly) should prevent the subsequent description of means and standard deviations to 1/1000 of an hour. A similar issue arises with the clinical scores where the ordinal scoring system of 0, 1, 2, 3, 4 is used, but the summary statistics use means and standard deviations to 1/1000 of the score. The nonparametric nature of the data was however recognised by the choice of the Mann-Whitney U test to analyse these data. An alternative way to present the data summary should be considered.

Response: Thank you for your comments, all calculated data were reduced to 2 decimal places in the figures in text and supplementary file and not I have the opinion that it’s more use friendly the way you suggested. Originally, we just copied form statistical software.

In the comparison between two groups before treatment we proved that both grups were in similar condition and to exclude this kind of bias.

Considering distribution, we did normality check on temperature data and time to resolution. It appears that we indeed have normally distributed temperatures, but our time to resolution in hours don’t conform the normal distribution. (as presented below)

Test of Normality (Shapiro-Wilk)

W

p

Temperature (°C)

Control

0.957

0.487

N-acetylcysteine in Tretment

0.940

0.239

Time to resolution (h)

Control

0.922

0.110

N-acetylcysteine in Tretment

0.886

0.023

Note. Significant results suggest a deviation from normality.

Test of Equality of Variances (Levene's)

F

df1

df2

p

Temperature (°C)

0.014

1

38

0.906

Time to resolution (h)

7.531

1

38

0.009

In further discussion we agreed that in general sense the temperature data (if only observed above physiological value) they don’t have to be normally distributed and that, indeed, in practice we see the most of the animals with low grade fever, therefore our data might be by chance normally distributed in our particular observation. Therefore, we have changed temperature and time to resolution statistical tests to nonparametric. Thank you again, we learned a lot from your comment.

Also, for our data in time to resolution we added both decimal and minute inscription for example that 41.4 hours is 41 hours and 24 minutes for calculated values, but we put in results as well that “In plain language, the NAC reduced therapy by one day.” Line 151-152

Results

Line 146: The word “symptom” is used incorrectly.

Response: We made a change with signs.

While the supplementary tables are referred to and summarised there is no mention of Figure 1 in the text, although the results of the t test are given.

Response: Please see the Line 149

Unclear in as much as there are 11 pink dots and 17 brown dots representing a total of 40 calves. Therefore each dot can presumably represent more than one calf.

Response: Thank you for your comments, after you pointed ad it, we tried to identify why, and in first we thought that there is some issue with the JASP software why it didn’t show all the dots, but after closer consideration and data sorting by temperature we have identified that several dots actually overlap. For example, three calves with 39.7 °C in NAC group had recovery period of 24 hours and those dots on diagram overlap, that’s why we see less than 20 dots by group.

Lines 166-167: These data show that on this occasion NAC reduced recovery  time by as much as one day and not that NAC reduces recovery time by 27  hours and 44 minutes in general.

Response: Thank you for comments. Those data were calculated, so we changed according to your suggestion that NAC reduced recovery time by one day, as it gives the practical meaning in every day usage.

4 Discussion

Lines 175-176: The purpose of this sentence is unclear and reference 24 is a literature review on diagnostic tests to detect pneumonia in pre-weaned dairy and veal calves and not a primary source on the occurrence of respiratory disease.

Response: Thank you for comments. We agree with you reference 24 not a primary source of data but, summarized data from multiple authors. We adde new reference.

Line 177-180: Sentence should be rewritten.

Response: English corrected.

Reviewer 2 Report

Comments and Suggestions for Authors

Thank you for the opportunity to review this article. This study aimed to investigate the effects of NAC in neonatal calves with respiratory disorders. The results indicated that the NAC treatment had significantly shorter time to resolution period. The quality of this manuscript is good. However, I have some questions as blow:

1. The enrollment/exclusion criteria was very important when the randomized control test was performed, so in this manuscript the enrollment/exclusion criteria of calves included in the study should be further clarified.

2. Why was amoxicilline with clavulanic acid chose for treatment in this experiment ?

3. What is the detail of the Wiskonsin clinical score? How to ensure that there was no significant difference between the scores of the two groups.

4. The clinical presentation was mainly assessed by visual inspection, how to limit the awareness of investigator to assessment assignment?

5. The results of clinical scores between compared groups before and after treatments should be listed in the manuscript.

Author Response

Dear reviewer

Firstly, we would like to thank you for your comments, as we see the way that they greatly improve our work. We tried to give adequate answers to your suggestions.

Comments

Thank you for the opportunity to review this article. This study aimed to investigate the effects of NAC in neonatal calves with respiratory disorders. The results indicated that the NAC treatment had significantly shorter time to resolution period. The quality of this manuscript is good. However, I have some questions as blow:

  1. The enrollment/exclusion criteria was very important when the randomized control test was performed, so in this manuscript the enrollment/exclusion criteria of calves included in the study should be further clarified.

Response: All calves were treated at the owner's request with a clinical picture of respiratory disease. based on the owner's decision to accept basic treatment or extended with NAC. With the clinical examination and the owner's decision to accept the treatment, the calves were evaluated and monitored during the therapy.

  1. Why was amoxicilline with clavulanic acid chose for treatment in this experiment?

Response: Thank you very much for this question as it’s raised among a couple of authors anyway a couple of years before. The answer is that is empirical knowledge and the initial price. Further on, the β lactam antibiotics are broad-spectrum antibiotics. Our experience has shown that the use of β-lactam antibiotics in newborn animals has successfully treated various disorders.

Similar investigation performed by Vaena et al., 2011:

Vaena, M. P., Sumathi, B. R.  Thearapeutic Management of Neonatal Calf Pneumonia in HF calf- A Case Report. Vet. World. 2011, 4(2

  1. What is the detail of the Wiskonsin clinical score? How to ensure that there was no significant difference between the scores of the two groups.

Response: Scoring and the specificity of the Wisconsin scoring system has been described by Vandermeulen J, Bahr C, Johnston D, Earley B, Tullo E, Fontana I, Guarino M, Exadaktylos V, Berckmans D. Early recognition of bovine respiratory disease in calves using automated continuous monitoring of cough sounds. Comput Electron Agric. 2016 Nov 1;129:15-26. doi: 10.1016/j.compag.2016.07.014.

In the comparison between two groups before treatment we proved that both groups were in similar condition and to exclude this kind of bias. The results of clinical scores between compared groups before treatments presented in the Supplementary file. We were thinking of adding the scores after treatment, but it’s nonsence as the calves were treated until recovery (so the parameters were all good after) and we have had no death outcomes

  1. The clinical presentation was mainly assessed by visual inspection, how to limit the awareness of investigator to assessment assignment?

For this reason we use Wisconsin score and in general practice, our senior member reminds us to regularly “speak in Wisconsin” and not in descriptive clinical terms. Furter more it was shown that temperature is the important factor of disease resolution time, and this parameter is objectively measured by rectal thermometer. We hope to reduce systemic error by using singular veterinarian to clinically check the status of calves, and that we have reduced error to 0.1°C.

  1. The results of clinical scores between compared groups before and after treatments should be listed in the manuscript.

Thank you for your comment, there were disagreement about this in the authors group as well. The most interesting data for wider public (someone is reading about NAC, somebody about calves’ respiratory diseases, somebody about recovery period) is the most statistically different. Therefore, we put focus in temperatures and time to resolution for both groups in the main text.

With initial comparations of clinical data showed that both of the groups are similar in their condition, so that we have uniform groups to start with.

Other data is interesting for us, who deal with cattle, about what scores we got on calves, how much those scores were different individually among both groups. We decided to present that data as well, but as supplementary where everybody can see but our main text is more focused on interventional study.

The data about linear regression is customed to be shown in the beginning and end of the analysis, the initial result is in supplementary and the final where we discuss the NAC supplement and it’s usefulness for calves, owners and us the vets, we put that “cream of the cake” in manuscript.

Kind regards,

Authors

Round 2

Reviewer 1 Report

Comments and Suggestions for Authors

Animals-3164843-peer-review-v2

General

The use of NAC as a supplementary treatment for calves suffering from pneumonia is a research area of value. The quality of the English has been greatly improved in this second version of the paper. Nevertheless the discussion, in particular, continues to lack focus and has too many unsubstantiated comments that do not appear to relate closely to this study. However the most important points remain: the calves were claimed to be randomly allocated to the two treatment groups and this is clearly not the case; all calves were assisted births, but no recognition was made of the severity of dystocia suffered; and no recognition was made of the impact that dystocia has on the chance the calf has of receiving sufficient immunoglobulin. As these important factors that impact on the health of the calves were not included in the study and because the allocation to the treatment groups was non-random the claim that the difference in recovery seen between the two treatments was the consequence of NAC treatment cannot be made.

Simple Summary: No comment

Abstract

Lines 20-22: The claim that “calves were randomly divided into two groups” is incorrect as the decision on whether calves were treated with NAC was ultimately made by the farmer and therefore NAC treated animals were not present on some of the farms. Where the farmer declined to pay for the NAC only control animals were present on the farm and therefore presumably calves in the treatment group were over-represented on some of the other farms.

Introduction

Line 30-31: This sentence is incorrect: calf pneumonia can occur at any time in the calfhood. References 1 and 2 do not appear to be studies on the aetiology of BRD.

Lines 31-33: This sentence is incorrect as it attributes the costs of BRD in cattle to BRD in neonatal calves and the claims of costs due to pneumonia in the neonatal calf is not supported by any reference.

Lines 33-34: This sentence is not related to the preceding or following text.

Line 37: Unnecessary word in “extremely lethal”: lethal is extreme enough. The risk of death “is due to the multifactorial etiology of relevant diseases,” is a dubious claim to make.

Lines 38-51: The discussion on the impact of dystocia on the health of the calf is insufficiently focussed.

Lines 49-50: Reference 8 is a case study with discussion on the nature of inhalation pneumonia in relation to the structure and function of the bovine lung. Likening amniotic fluid, which is naturally present in the respiratory tract as the calf is born to the “50 mls of aspirate” that can result in inhalation pneumonia is misleading. However, at no point in reference 8 is mention made of 50 ml of aspirate.

Lines 70-71: The claim is made that the study evaluated the mucolytic and anti-inflammatory activities of NAC. No evaluation of the mucolytic activity nor of the inflammatory response other than assessing the calves clinically was made. Therefore this claim is incorrect.

General: while much is made of the impact of dystocia on the calf nothing is mentioned about the transfer of passive immunity, which plays a critical role in the resistance of the calf to infection and its ability to maintain good health.

Materials and Methods

Line 80: Dystocia has a well-understood impact on the chance of survival for the calf. The degree of dystocia suffered can be expected to vary between assisted calvings. No assessment of the degree of dystocia has been made. This is therefore an important variable that has not been accounted for.

Line 81: Would be simpler to write, “no calf born by c-section was included in the study”.

Line 84-86 and Lines 91-94: There is some repetition between these two sections. The methodology is not adequately explained as the word random is used,  but the process is not detailed and the role that the farmer has in determining whether or not the calf received NAC removes any possibility of random allocation to treatment or control groups. This is a serious flaw in the study. Ultimately any difference between the control and treatment groups may have been due to a farm/calf carer effect and not to the treatment.

Lines 89-90: It is difficult to understand the relevance of ref 19 in relation to selection of cases. It is sufficient to cite the Wisconsin scoring system.

Lines 117-132: Despite stating after this that the rectal temperature was measured twice a day, it is not clear how frequently the calves were examined by a veterinary clinician and assessed using the Wisconsin scoring system. While it states in line 130 that “Calves followed a morning-evening schedule during the treatment period” this provides insufficient detail to understand how frequently the calves were examined and by whom.

Line 118: Recovered would be a better word than cured.

Lines 134-137: The clinical variables (ocular and nasal discharge, cough and ear position) were ordinal and not continuous and therefore the descriptive statistics are not an adequate way to examine and present these data. These combined together allow the clinician to assess whether the calf had recovered, and it is the time to recovery that is the critical evaluation.

Results

The supplementary tables have not been completely translated into English.

Figure 1: The description of the graphic is inadequate as no explanation is offered as to why there are fewer dots than calves and all dots are the same size and only have different shades of grey between the treatment and control groups. Therefore dots for one or more than one calf cannot be differentiated from each other, and it is therefore not possible to interpret the graphic.

Lines 148 to 153: The use of the mean to describe the time to recovery for the two treatment groups is questionable as it is treating the time as a continuous measured variable, but all that is known is that the calves had their rectal temperature taken twice a day and the time between examinations is not detailed. Assuming that the calves were indeed examined and assessed by the Wisconsin scoring system twice a day a more appropriate assessment would be the number of examination points made before recovery was achieved. The assessment of difference between the groups would then be the median number of examinations to recovery and the range could be given by the quartiles and the minimum and maximum.

Line 155 to 159: The linear regression model did not include farm as a dependent variable and as explained previously, this could have had an important effect on time to recovery.

Discussion

Lines 174-176: Ref 24 is the NAHMS study on dairy heifer health that involved 104 dairy operations in the USA. 33.4% of the pre-weaned calves were treated for respiratory signs and the peak age for this condition was five weeks of age. That is the reference does not support the claim.

Line 177: The word “symptoms” is used incorrectly.

Line 193: Ref 5 (NAHMS study of factors associated with mortality to 21 days of life from 829 dairy operations in USA) and ref 31 (a study of calves born prematurely) are not studies on the initiation of pneumonia in newborn calves and do not support the text.

Lines 193-195: This is a dubious statement and the reference to support it (26) could not be found.

Lines 213: (CVMP) should be moved to follow Commission for Veterinary Medicinal Products in the preceding line.

Lines 213 to 216: This statement requires to be supported with a reference.

Lines 220-222: The three references quoted are not studies on the trend of the use of NAC as a therapeutic agent as claimed.

Lines 230-2331: The claim that “maintaining the antioxidant status of goats (whatever that means) can improve the quality of milk and the health status of milking goats requires to be supported by a reference.

Lines 249-256: It is difficult to understand why this has been included.

Line 260: Respiratory distress syndrome is used for the first and only time in the text in the discussion of the use of NAC. This is confusing. This term is mostly applied to premature calves that are unable to breathe properly and is not related to pneumonia.

Lines 292-294: Most of this text is difficult to follow and is not supported by a reference.

Comments on the Quality of English Language

Moderate editing of English language required.

Author Response

Dear reviewer

Firstly, we would like to thank you for your comments, as we see the way that they greatly improve our work. The final version of the manuscript was revised over MDPI language editing service.

General

The use of NAC as a supplementary treatment for calves suffering from pneumonia is a research area of value. The quality of the English has been greatly improved in this second version of the paper. Nevertheless the discussion, in particular, continues to lack focus and has too many unsubstantiated comments that do not appear to relate closely to this study. However the most important points remain: the calves were claimed to be randomly allocated to the two treatment groups and this is clearly not the case; all calves were assisted births, but no recognition was made of the severity of dystocia suffered; and no recognition was made of the impact that dystocia has on the chance the calf has of receiving sufficient immunoglobulin. As these important factors that impact on the health of the calves were not included in the study and because the allocation to the treatment groups was non-random the claim that the difference in recovery seen between the two treatments was the consequence of NAC treatment cannot be made.

Simple Summary: No comment

Abstract

Lines 20-22: The claim that “calves were randomly divided into two groups” is incorrect as the decision on whether calves were treated with NAC was ultimately made by the farmer and therefore NAC treated animals were not present on some of the farms. Where the farmer declined to pay for the NAC only control animals were present on the farm and therefore presumably calves in the treatment group were over-represented on some of the other farms.

Response:  Thank you for comment. We deleted the claim “randomly” from manuscript. About randomly allocated calves, we took as a farmers free will to participate or not in the NAC treatment according to his previous experience. We are aware that it would be much statistically better if there was both C and NAC calves in every farm, more over to be put in the same stall Our study is of observational nature from small farm holders, and not from experimental farm.  

Introduction

Line 30-31: This sentence is incorrect: calf pneumonia can occur at any time in the calfhood. References 1 and 2 do not appear to be studies on the aetiology of BRD.

Response: We make changes in the sentence and have performed a references replacement.

  1. Maier GU, Love WJ, Karle BM, Dubrovsky SA, Williams DR, Champagne JD, Anderson RJ, Rowe JD, Lehenbauer TW, Van Eenennaam AL, Aly SS. Management factors associated with bovine respiratory disease in preweaned calves on California dairies: The BRD 100 study. J Dairy Sci. 2019 Aug;102(8):7288-7305.
  2. Studer, E., Schönecker, L., Meylan, M., Stucki, D., Dijkman, R., Holwerda, M., Glaus, A., Becker, J. Prevalence of BRD-Related Viral Pathogens in the Upper Respiratory Tract of Swiss Veal Calves. Animals. 2021, J29;11(7):1940.

Lines 31-33: This sentence is incorrect as it attributes the costs of BRD in cattle to BRD in neonatal calves and the claims of costs due to pneumonia in the neonatal calf is not supported by any reference.

Response: We make changes in the sentence and have performed a reference replacement.

Dubrovsky SA, Van Eenennaam AL, Aly SS, Karle BM, Rossitto PV, Overton MW, Lehenbauer TW, Fadel JG. Preweaning cost of bovine respiratory disease (BRD) and cost-benefit of implementation of preventative measures in calves on California dairies: The BRD 10K study. J Dairy Sci. 2020 Feb;103(2):1583-1597.

Lines 33-34: This sentence is not related to the preceding or following text.

Response: Thank you for comment. We deleted the sentence.

Line 37: Unnecessary word in “extremely lethal2: lethal is extreme enough. The risk of death “is due to the multifactorial etiology of relevant diseases,” is a dubious claim to make.

Response: Thank you for comment. We accepted your recommendation and make changes.

Lines 38-51: The discussion on the impact of dystocia on the health of the calf is insufficiently focussed.

Response: Thanks for the comment. We have modified the sentences and added two new references.  Effect of dystocia on health, survival and passive immunoglobulin transfer of calves.

Lombard, J. E., Garry, F. B., Tomlinson, S. M., & Garber, L. P. (2007). Impacts of dystocia on health and survival of dairy calves. Journal of dairy science, 90(4), 1751-1760.

Murray CF, Veira DM, Nadalin AL, Haines DM, Jackson ML, Pearl DL, Leslie KE. The effect of dystocia on physiological and behavioral characteristics related to vitality and passive transfer of immunoglobulins in newborn Holstein calves. Can J Vet Res. 2015 Apr;79(2):109-19

Lines 49-50: Reference 8 is a case study with discussion on the nature of inhalation pneumonia in relation to the structure and function of the bovine lung. Likening amniotic fluid, which is naturally present in the respiratory tract as the calf is born to the “50 mls of aspirate” that can result in inhalation pneumonia is misleading. However, at no point in reference 8 is mention made of 50 ml of aspirate.

Response: Thank you for comment. We deleted the claim from the manuscript.

Lines 70-71: The claim is made that the study evaluated the mucolytic and anti-inflammatory activities of NAC. No evaluation of the mucolytic activity nor of the inflammatory response other than assessing the calves clinically was made. Therefore this claim is incorrect.

Response: Thanks for the comment. We deleted the claims from the manuscript.

General: while much is made of the impact of dystocia on the calf nothing is mentioned about the transfer of passive immunity, which plays a critical role in the resistance of the calf to infection and its ability to maintain good health.

Response: Thank you for your comment, we have supplemented the work according to the above-mentioned comment.

Materials and Methods

Line 80: Dystocia has a well-understood impact on the chance of survival for the calf. The degree of dystocia suffered can be expected to vary between assisted calvings. No assessment of the degree of dystocia has been made. This is therefore an important variable that has not been accounted for.

Response: Thank you for comment. We couldn’t agree more with you, however we don’t have data to assessment of the degree of dystocia has been made. We have considered only dystocia history in deliveries as anamnestic data for calves, the obstetric procedures were done by several different veterinarians and specialistic services chosen by the farmer. The closer introduction to delivery pathology is not in the scope of our work, and I am afraid, that not all the doctors would be willing to share data.

Line 81: Would be simpler to write, “no calf born by c-section was included in the study”.

Response: Thank you for comment. We accepted your recommendation. This way the manuscript is really more “reader friendly”.

Line 84-86 and Lines 91-94: There is some repetition between these two sections. The methodology is not adequately explained as the word random is used,  but the process is not detailed and the role that the farmer has in determining whether or not the calf received NAC removes any possibility of random allocation to treatment or control groups. This is a serious flaw in the study. Ultimately any difference between the control and treatment groups may have been due to a farm/calf carer effect and not to the treatment.

Response: Thank you for your comment, we are aware that there might be a design flaw because the study has been done on small holdings and we haven’t had control of putting calves to same condition. If we had the opportunity to conduct controlled environment experiment, we would put each C and NAC calves in the same dedicated stall on the single farm to minimise the eventual care bias. In order to control the care factor we have done 2x time a day health check. Also, we have added care taking observation sentence in lines 358-359 in discussion. All farms have similar housing conditions and feeding method, they are within 20 km and we don’t expect much influence of different farm management. But it certainly stands your point, and for some other observations the farm micro locations and management could make a huge difference.

Lines 89-90: It is difficult to understand the relevance of ref 19 in relation to selection of cases. It is sufficient to cite the Wisconsin scoring system.

Response: We deleted the ref 19.

Lines 117-132: Despite stating after this that the rectal temperature was measured twice a day, it is not clear how frequently the calves were examined by a veterinary clinician and assessed using the Wisconsin scoring system. While it states in line 130 that “Calves followed a morning-evening schedule during the treatment period” this provides insufficient detail to understand how frequently the calves were examined and by whom.

Response: Thank you we have widened this by explaining that health check was done twice a day for all affected calves.

Line 118: Recovered would be a better word than cured.

Thank you, indeed we agree, and we have changed the term to cured

Lines 134-137: The clinical variables (ocular and nasal discharge, cough and ear position) were ordinal and not continuous and therefore the descriptive statistics are not an adequate way to examine and present these data. These combined together allow the clinician to assess whether the calf had recovered, and it is the time to recovery that is the critical evaluation.

Response: We have put additional median, interquartile range, minimum and maximum (thus the range) in the supplementary files.

Results

The supplementary tables have not been completely translated into English.

Response: We made changes in the Supplementary fille.

Figure 1: The description of the graphic is inadequate as no explanation is offered as to why there are fewer dots than calves and all dots are the same size and only have different shades of grey between the treatment and control groups. Therefore dots for one or more than one calf cannot be differentiated from each other, and it is therefore not possible to interpret the graphic.

Response: Thank you for your comments, after you pointed ad it, we tried to identify why, and in first we thought that there is some issue with the JASP software why it didn’t show all the dots, but after closer consideration and data sorting by temperature we have identified that several dots actually overlap. For example, three calves with 39.7 °C in NAC group had recovery period of 24 hours and those dots on diagram overlap, that’s why we see less than 20 dots by group. We also wrote an email to JASP team and they responded that we only can make jitter to our graph. In this new graph there is no trend line between temperature and time to resolution seen for our readers. 

Graph type 1

Graph type 2

Lines 148 to 153: The use of the mean to describe the time to recovery for the two treatment groups is questionable as it is treating the time as a continuous measured variable, but all that is known is that the calves had their rectal temperature taken twice a day and the time between examinations is not detailed. Assuming that the calves were indeed examined and assessed by the Wisconsin scoring system twice a day a more appropriate assessment would be the number of examination points made before recovery was achieved. The assessment of difference between the groups would then be the median number of examinations to recovery and the range could be given by the quartiles and the minimum and maximum.

Response: The scale is there to easy examination process, unify and quantify clinical signs in calves the rectal temperature monitoring as the step that require the most handling of the calves during examination. The experienced clinician can score parameters Wisconsin scoring system in a matter of minutes, the eye/nasal discharge, ear position, fecal consistency, cough (even with inducing the cough). Moreover this calves needed auscultation for health assessment. The assessment between groups we did the regression analysis which is a far more informative than the mean. We got the result that 27.74 hours (27 hours and 44 minutes), of recovery time is shortened if NAC is used. We did analysis in the hours, as the hours would be universal method of measurement, if we in the examination points it would be 2.3116. The conclusion remains the same, for about one day or two veterinarian visits, the period for clinical signs resolution is shortened if NAC is used. The sense of time is more applicable if spoken in hours than in visits, because for some diseases the visits are once daily, or every other day, or else.

Line 155 to 159: The linear regression model did not include farm as a dependent variable and as explained previously, this could have had an important effect on time to recovery.

Response: Thank you very much, all calves comes from 4 farms.

Discussion

Lines 174-176: Ref 24 is the NAHMS study on dairy heifer health that involved 104 dairy operations in the USA. 33.4% of the pre-weaned calves were treated for respiratory signs and the peak age for this condition was five weeks of age. That is the reference does not support the claim.

Response: We have changed the references with reference:    Donlon, J. D., Mee, J. F., & McAloon, C. G. (2023). Prevalence of respiratory disease in Irish preweaned dairy calves using hierarchical Bayesian latent class analysis. Frontiers in Veterinary Science, 10, 1149929.

Line 177: The word “symptoms” is used incorrectly.

Response: use of the word “symptom” and it should be clinical sign 

Response: We accepted your recommendation and changes.  

Line 193: Ref 5 (NAHMS study of factors associated with mortality to 21 days of life from 829 dairy operations in USA) and ref 31 (a study of calves born prematurely) are not studies on the initiation of pneumonia in newborn calves and do not support the text.

Lombard, J. E., Garry, F. B., Tomlinson, S. M., & Garber, L. P. (2007). Impacts of dystocia on health and survival of dairy calves. Journal of dairy science, 90(4), 1751-1760.

Lines 193-195: This is a dubious statement and the reference to support it (26) could not be found.

Response:  We make change in the sentence and replacement of reference:

Binversie, E. S., Ruegg, P. L., Combs, D. K., & Ollivett, T. L. (2020). Randomized clinical trial to assess the effect of antibiotic therapy on health and growth of preweaned dairy calves diagnosed with respiratory disease using respiratory scoring and lung ultrasound. Journal of Dairy Science, 103(12), 11723-11735.

Lines 213: (CVMP) should be moved to follow Commission for Veterinary Medicinal Products in the preceding line.

Response: We accepted your recommendation.

Lines 213 to 216: This statement requires to be supported with a reference.

Response: We added new reference:

Schwalfenberg GK. N-Acetylcysteine: A Review of Clinical Usefulness (an Old Drug with New Tricks). J Nutr Metab. 2021 Jun 9; 2021:9949453.

Lines 220-222: The three references quoted are not studies on the trend of the use of NAC as a therapeutic agent as claimed.

Response: We replacement the references:

Caissie, M. D., Gartley, C. J., Scholtz, E. L., Hewson, J., Johnson, R., Chenier, T. The effects of treatment with N-acetyl cysteine on clinical signs in persistent breeding-induced endometritis susceptible mares. JEVS, 2020, 92, 103142

Wang, L., Zhou, J., Hou, Y., Yi, D., Ding, B., Xie, J., Zhang, Y., Chen, H., Wu, T., Zhao, D., Hu, C-A.A., Wu, G. N-Acetylcysteine supplementation alleviates intestinal injury in piglets infected by porcine epidemic diarrhea virus. Amino Acids, 2017, 49, 1931-1943

Kang, K. S., Shin, S., & Lee, S. I. N-acetylcysteine modulates cyclophosphamide-induced immunosuppression, liver injury, and oxidative stress in miniature pigs. JAST. 2020, 62(3),

Lines 230-231: The claim that “maintaining the antioxidant status of goats (whatever that means) can improve the quality of milk and the health status of milking goats requires to be supported by a reference.

Response: Thank you for comments, we added reference.

Jóźwik, A., Bagnicka, E., StrzaÅ‚kowska, N., Åšliwa-Jóźwik, A., HorbaÅ„czuk, K., Cooper, R. G., ... & HorbaÅ„czuk, J. O. (2010). The oxidative status of milking goats after per os administration of N-acetylcysteine. Animal Science Papers and Reports, 28(2), 143-152.

Lines 249-256: It is difficult to understand why this has been included.

Response: These few sentences form the basis for further discussion. Several papers have described the potential use of NAC, which exhibits antimicrobial effects.

Line 260: Respiratory distress syndrome is used for the first and only time in the text in the discussion of the use of NAC. This is confusing. This term is mostly applied to premature calves that are unable to breathe properly and is not related to pneumonia.

Response: We have wrote pneumonia in manuscript because the pneumonia is diagnosed based on the following clinical protocol: Temperature elevated, respiratory clinical signs and Wisconsin scores, nasal discharge, additional respiratory auscultation. AS the lung auscultation is not the part of the Wisconsin clinical scoring system, the data is not shown.

Lines 292-294: Most of this text is difficult to follow and is not supported by a reference.

Response: We rearranged the sentence and added supported by a reference reference.

Reviewer 2 Report

Comments and Suggestions for Authors

I have no additional comments and suggestions.

Author Response

Thank you for comments , as we see the way that they greatly improve our work.

Authors